# Modulation of the Respiratory Epithelium Physiology by Flavonoids—Insights from 16HBEσcell Model

**DOI:** 10.3390/ijms252211999

**Published:** 2024-11-08

**Authors:** Jakub Hoser, Gabriela Weglinska, Aleksandra Samsel, Kamila Maliszewska-Olejniczak, Piotr Bednarczyk, Miroslaw Zajac

**Affiliations:** Department of Physics and Biophysics, Institute of Biology, Warsaw University of Life Sciences, 02-776 Warsaw, Poland; jakub_hoser@sggw.edu.pl (J.H.); s205260@sggw.edu.pl (G.W.); aleks.samsel@gmail.com (A.S.); kamila_maliszewska-olejniczak@sggw.edu.pl (K.M.-O.); miroslaw_zajac@sggw.edu.pl (M.Z.)

**Keywords:** airway epithelium, flavonoids, transepithelial electrical resistance, wound healing

## Abstract

Extensive evidence indicates that the compromise of airway epithelial barrier function is closely linked to the development of various diseases, posing a significant concern for global mortality and morbidity. Flavonoids, natural bioactive compounds, renowned for their antioxidant and anti-inflammatory properties, have been used for centuries to prevent and treat numerous ailments. Lately, a growing body of evidence suggests that flavonoids can enhance the integrity of the airway epithelial barrier. The objective of this study was to investigate the impact of selected flavonoids representing different subclasses, such as kaempferol (flavonol), luteolin (flavone), and naringenin (flavanone), on transepithelial electrical resistance (TEER), ionic currents, cells migration, and proliferation of a human bronchial epithelial cell line (16HBE14σ). To investigate the effect of selected flavonoids, MTT assay, trypan blue staining, and wound healing were assessed. Additionally, transepithelial resistance and Ussing chamber measurements were applied to investigate the impact of the flavonoids on the electrical properties of the epithelial barrier. This study showed that kaempferol, luteolin, and naringenin at micromolar concentrations were not cytotoxic to 16HBE14σ cells. Indeed, in MTT tests, a statistically significant change in cell metabolic activity for luteolin and naringenin was observed. However, our experiments showed that naringenin did not affect the proliferation of 16HBE14σ cells, while the effect of kaempferol and luteolin was inhibitory. Moreover, transepithelial electrical resistance measurements have shown that all of the flavonoids used in this study improved the epithelial integrity with the slightest effect of kaempferol and the significant impact of naringenin and luteolin. Finally, our observations suggest that luteolin increases the Cl- transport through cystic fibrosis transmembrane conductance regulator (CFTR) channel. Our findings reveal that flavonoids representing different subclasses exert distinct effects in the employed cellular model despite their similar chemical structures. In summary, our study sheds new light on the diverse effects of selected flavonoids on airway epithelial barrier function, underscoring the importance of further exploration into their potential therapeutic applications in respiratory health.

## 1. Introduction

The epithelium lines all moist surfaces of the human body, including the intestines, secretory organs, and the reproductive and respiratory tracts, performing various functions such as secretion, absorption, excretion, filtration, sensory reception, and protection [1,2]. The respiratory epithelium is an interface between the internal milieu and the external environment, continuously exposed to inhaled allergens, particles, and pathogens, which must be cleared without triggering inflammation to maintain homeostasis [3,4]. To prevent environmental triggers and microbes from penetrating the submucosa, the epithelium employs three fundamental mechanisms: mucin secretion to trap pollutants, production of antimicrobial peptides and defensins to inhibit microbial growth, and the formation of a tight physical barrier to protect underlying tissues and prevent the infiltration of potential threats [5]. The airway barrier can be disrupted by increased permeability, impaired mucociliary clearance, chronic inflammation, or oxidative stress [6]. Substantial evidence shows that a compromised airway epithelial barrier is closely associated with the development of diseases such as asthma, chronic obstructive pulmonary disease, and cystic fibrosis, which significantly contribute to global mortality and morbidity [7,8,9,10,11]. Current research is focused on treatments that target airway barrier dysfunction, including anti-inflammatory therapies to reduce inflammation, interventions to enhance epithelial barrier integrity, antioxidant use to mitigate oxidative stress, and mucolytics to improve mucociliary clearance [12,13].

Flavonoids are a diverse group of plant-derived secondary metabolites that belong to a larger class of polyphenolic compounds [14]. They are widely found in nuts, vegetables, and fruits [15]. So far, more than 10,000 different flavonoid compounds have been found, extracted, isolated and identified [16], and new structures are being reported every year [17]. Flavonoids share a standard chemical structure. The core consists of two phenyl and one heterocyclic ring. Depending on the chemical structure, degree of saturation, and the oxidation of the carbon ring, flavonoids are classified into one of seven significant subclasses such as flavanones, flavones, isoflavones, flavonols, flavanonols, flavanols, and anthocyanins [18].

Due to their broad therapeutic effects, flavonoids have attracted increasing research interest in recent years, especially in areas such as inflammation, oxidative stress, cancer, cardiovascular diseases, and neurodegenerative conditions [19]. The vast diversity of flavonoid chemical structures results in a large array of their biological effects [20] might exert beneficial effects on airway epithelium in health and disease. Studies have shown that a diet rich in flavonoids correlates with improved lung function and a reduction in chronic respiratory diseases (see review [21]). Most flavonoids do not harm normal epithelial cells [22] but exhibit selective toxicity to cancer cells by increasing intracellular reactive oxygen species (ROS) levels. Previous research also suggests that flavonoids can improve airway barrier integrity and support the proliferation of respiratory epithelial cells [23,24]. Additionally, flavonoids have been shown to modulate ion-transporting proteins critical for normal lung function, such as the epithelial sodium channel (ENaC) [25], cystic fibrosis transmembrane conductance regulator (CFTR) [26,27], and calcium-dependent chloride channels (CaCC) [28].

Interest in the use of flavonoids for respiratory health is growing significantly. Recent studies show their potential as natural therapeutic agents for respiratory diseases such as chronic obstructive pulmonary disease, asthma, acute lung injury, acute respiratory distress syndrome, and respiratory infections. While there is currently no treatment for chronic respiratory diseases, developing new drugs is in great demand. The objective of this study was to investigate the impact of selected flavonoids such as kaempferol, luteolin, and naringenin on various aspects of airway epithelium parameters such as transepithelial electrical resistance (TEER), ionic currents, wound-healing properties, and proliferation. Our results indicate that these flavonoids exert distinct effects, despite their structural similarities, underscoring their diverse impacts on airway barrier function and highlighting the need for further exploration of their potential in respiratory health applications.

## 2. Results

### 2.1. Flavonoids Affect Cell Metabolic Activity but Not Cell Viability

The results of the MTT assay and trypan blue staining are presented in Figure 1A and Figure 1B, respectively. The changes in metabolic activity were observed for all tested flavonoids. Incubation of cells with kaempferol for 24 h led to non-significant changes in the drop of metabolic activity when used at 10 μM and 50 μM. However, the rise was observed for 30 μM. Incubation with luteolin and naringenin caused statistically significant changes in metabolic activity when used at 10 and 30 μM concentrations. No significant changes were observed for both flavonoids when used at the highest concentration tested (50 μM).

The trypan blue staining assesses the possible cytotoxicity of selected flavonoids. As presented in Figure 1B, none of the flavonoids tested had a cytotoxic effect on 16HBE14σ cells.

### 2.2. Flavonoids Exert Different Effects on Cell Migration

The effect of flavonoids on cell migration/proliferation was assessed by scratch assay. Figure 2 presents the scratch area changes in the presence and absence (control) of flavonoids. The effect of kaempferol and luteolin was observed, while no changes were detected for cells incubated with naringenin. A small, however statistically significant, concentration-dependent inhibitory effect was observed for kaempferol at all the concentrations used—Figure 2A. Luteolin had a strong and concentration-dependent inhibitory effect on cell proliferation—Figure 2B. At the highest concentration used (50 μM), cell proliferation was nearly entirely suppressed. On the other hand, naringenin did not affect the kinetics of scratch closure—Figure 2C. The rate of scratch closure at different time intervals was presented for control and luteolin in Figure 2D.

### 2.3. Flavonoids Affect the Transepithelial Electrical Resistance

After assessing flavonoids’ effects at the cellular level, a series of electrophysiological experiments were conducted. The first step was to evaluate the impact of flavonoids on epithelial barrier integrity by the measurement of TEER. Medium supplemented with various concentrations of kaempferol, luteolin, and naringenin was applied to the apical side of the cell monolayer, and TEER was measured after 24, 48, and 72 h. The TEER values over time are presented in Figure 3. All tested flavonoids increased TEER, with kaempferol showing the most minor effect (Figure 3A). Luteolin and naringenin produced a significant TEER increase in a concentration-dependent manner (Figure 3B and Figure 3C, respectively), with the highest increase observed at concentrations of 30 and 50 μM for both flavonoids.

### 2.4. Ussing Chamber Measurement

Based on all of the experiments performed, luteolin was selected for the ion transport measurements across the cell monolayers. Polarized cell monolayers were mounted in EasyMount Ussing chamber system (Physiologic Instruments, Reno, NV, USA), basolateral to apical chloride gradient was applied across the monolayer, and the short circuit current was measured under voltage-clamp configuration. Experiments were performed on luteolin-treated (50 μM, 72 h) and non-treated cell monolayers.

Luteolin increased the basal current passing through cell monolayers (Figure 4). To further analyze the effect of luteolin treatment on the cystic fibrosis transmembrane conductance regulator (CFTR) channel activity, the cAMP-dependent Cl^−^ transport was assessed by the addition of forskolin (Figure 5A,B) and the subsequent addition of CFTR_inh_-172. In cell monolayers treated with luteolin, the effect of forskolin and CFTR_inh_-172 was also more substantial compared to the control (Figure 5A,B). Neither the calcium-dependent Cl^−^ transport induced by ATP nor the Na^+^ transport assessed by the amiloride were modified by luteolin (Figure 5A,B).

## 3. Discussion

Recent studies indicate that respiratory disorders are among the primary causes of mortality and morbidity worldwide [29,30]. The airway epithelium protects us from environmental threats, thus maintaining airway barrier function is vital for respiratory health. The barrier is formed by epithelial junctions restricting permeability to inhaled environmental stressors and pathogens. The airway epithelium barrier disruption exposes subepithelial layers to hazardous agents present in inhaled air but also alters their normal function by the modulation of signaling pathways involved in repair, differentiation, and proinflammatory responses [31], leading to significant pathophysiological consequences. Thus, targeting epithelial barrier defects is an attractive therapeutic strategy [31].

Flavonoids hold significant, multi-faceted potential for improving airway barrier function, offering protective and therapeutic benefits for respiratory health. Clinical studies have shown that flavonoid intake increases pulmonary function [32,33]. However, their effects on airway epithelial barrier properties remain incomplete. In this study, we focused on the potential benefits of flavonoids representing different subclasses: kaempferol (flavonol), luteolin (flavon), and naringenin (flavanone) on the function of the respiratory epithelium. Our results show that these flavonoids, despite similar structures, have distinct effects on the 16HBE14σ cell line.

Kaempferol, luteolin, and naringenin at the concentrations used in our experiments were not cytotoxic to 16HBE14σ cells, as shown by trypan blue staining. This is in agreement with literature describing most flavonoids as selectively toxic for cancer but not for normal cells [34,35]. Indeed, in MTT tests, the statistically significant change in cell metabolic activity for luteolin and naringenin was observed; however, this might not be due to the cellular effect, but the methodology used. Several studies highlight the limitations of the MTT assay for assessing cell viability, toxicity, and proliferation in flavonoid testing. Previous research has shown that flavonoids may affect MTT reduction even in the absence of cells [36,37,38]. Somayaji and Shastry [38] reported reversed MTT assay profiles starting at flavonoid concentrations of 25 μM. In our study, we observed a decrease in cell metabolic activity for all tested flavonoids at the lowest concentrations, though no clear dose–response relationship was evident for luteolin and kaempferol. Several factors may explain these observations: the lack of a time- and concentration-dependent trend for some flavonoids, as previously noted by Talorete et al. [37]; differences in membrane permeability and cellular accumulation of flavonoids [39]; flavonoid stability in cell culture medium; and potentially distinct bioactivities of metabolized or degraded flavonoids. Although we did not observe a dose–response relationship, we found no statistically significant changes in cell metabolic activity across different flavonoid concentrations.

Several studies have demonstrated the significance of flavonoids as agents promoting wound healing in various epithelial tissues [40,41,42]. However, their effects on airway epithelium, especially normal epithelium, have not been investigated in depth. Flavonoids were also shown to improve the transepithelial barrier integrity. Kaempferol, luteolin, and naringenin have been described to decrease the cell migration of cancer cells, making them promising agents in cancer treatment.

Our experiments demonstrated that naringenin did not affect the proliferation of 16HBE14o- cells; however, it significantly improved epithelial barrier integrity. Studies on other bronchial epithelial models, such as normal human bronchial epithelium (NHBE) and BEAS-2B cells, have shown that naringenin reduces proinflammatory cytokine secretion and decreases intracellular reactive oxygen species levels, suggesting a primarily protective role rather than a direct proliferative effect. These findings align with our results from the 16HBE14o- cell model [23,24].

Kaempferol and luteolin exerted a concentration-dependent inhibitory effect on cell proliferation while increasing epithelial integrity. Transepithelial electrical resistance (TEER) measurements indicated that all flavonoids in this study improved epithelial integrity, with kaempferol showing the smallest (non-significant) effect, while naringenin and luteolin had a more substantial impact. Similar effects have been observed in the intestinal epithelium [43,44,45]. Kaempferol, in particular, appears to enhance airway epithelial integrity by modulating signaling pathways and inflammatory responses. Studies using the BEAS-2B cell model have shown that kaempferol reduces the expression of nicotinamide adenine dinucleotide phosphate oxidase 4 (NOX4) and suppresses TNF-α-induced airway inflammation [22,46]. Kaempferol also positively affects mucus production by reducing MUC5AC protein levels [47]. Observations from our study and others, as reviewed by Mishra et al. [48], suggest that kaempferol holds substantial promise for treating airway diseases such as asthma and chronic obstructive pulmonary disease (COPD).

Luteolin has been described to inhibit the proliferation of various cancer cell lines [49], though its effects on normal, wild-type cells, especially respiratory epithelial cells, are less well known. Our results show that luteolin inhibits the proliferation of the 16HBE14o- cell line in a concentration-dependent manner, with proliferation almost completely halted at the highest concentration (50 μM). Simultaneously, epithelial barrier integrity, assessed by TEER measurements, improved. While the direct effects of luteolin on TEER in bronchial cells have not been explicitly detailed, previous studies suggest that luteolin’s positive impact may be related to its ability to reduce oxidative stress and inflammation [50,51].

The combined observation of inhibited cell proliferation and increased TEER for kaempferol and luteolin suggests that these flavonoids enhance barrier integrity in 16HBE14o- cells, likely through the upregulation of tight junction expression, similar to findings in the intestinal Caco-2 model [44,52].

Flavonoids were also shown to affect the activity of ion-transporting proteins present in epithelium. For example, apigenin, genistein, and kaempferol already have been described for their potential to activate chloride currents through the CFTR channel [53]. Due to the lack of information about the possible effects of luteolin on chloride transport across airway epithelia, short-circuit current measurements were performed. The cell monolayers treated with luteolin at 50 μM for 72 h had higher basal currents compared to non-treated cell monolayers. Additionally, the response to forskolin and subsequently added CFTR_Inh_-172 were more potent in the cells treated with luteolin. These observations suggest that luteolin increases the Cl^−^ transport through CFTR; however, the mechanism of action needs to be found. The possible mechanism could include a direct activatory effect on the CFTR channel, but also the activation of basolateral NaK2Cl cotransporter and/or basolateral potassium channels known to increase the driving force for chloride efflux at the apical side of the epithelium [54]. This activation opposes the findings from T84 human colon epithelia, where acute luteolin addition had an inhibitory effect on CFTR chloride currents [55]. An increase in ENaC channel activity in luteolin-treated cells was also reported in the lung alveolar model [51]. Indeed, we observed an increased response to amiloride in luteolin-treated cells; however, this was marginal. No effect of treatment on purinergic receptors was observed, indicating that luteolin affects mainly the CFTR channel activity. However, the findings from Calu-3 cell models have shown that luteolin exhibits an inhibitory effect on calcium-dependent chloride channels (CaCC) [56]. The reason for this discrepancy might be explained by the different cell models used in both studies.

Among the flavonoids tested, luteolin shows great potential for treating chronic respiratory diseases such as asthma and acute respiratory distress syndrome. Its ability to increase transepithelial electrical resistance in bronchial epithelium could help protect underlying tissues from environmental threats present in inhaled air. Additionally, by reducing inflammation and oxidative stress, luteolin may provide therapeutic benefits for individuals with chronic obstructive pulmonary disease (COPD). Moreover, luteolin increased chloride currents in bronchial epithelium, suggesting a potential positive effect on airway surface liquid homeostasis. This could be particularly beneficial for individuals with cystic fibrosis, though further experiments are needed to explore this potential.

### Study Limitations

While this study provides valuable insights into the effects of selected flavonoids on airway epithelial barrier function, several limitations should be acknowledged. First, the study was conducted using a human bronchial epithelial cell line (16HBE14σ), which, although a widely used model, may not fully replicate the complexity of in vivo human airway conditions. The influence of factors such as the presence of immune cells, mechanical stress from breathing, and interactions with other cell types in the respiratory system were not accounted for, potentially limiting the generalizability of the findings.

Secondly, only three flavonoids were examined, representing different subclasses. The results may not necessarily extend to other flavonoids, and the specific mechanisms by which each compound influences epithelial barrier function remain unclear. Also, flavonoids are commonly transformed by metabolic processes, and perhaps the results from in vitro experiments should not be directly translated into in vivo systems. Further research is needed to investigate a broader range of flavonoids and clarify their molecular mechanisms of action.

Lastly, while TEER, ionic currents, wound-healing properties, and cell proliferation were evaluated, other aspects of airway epithelial integrity, such as mucus production, response to pathogens, and long-term barrier maintenance, were not explored. Future studies incorporating these factors would provide a more comprehensive understanding of flavonoid effects on respiratory health.

## 4. Materials and Methods

### 4.1. Flavonoids

Kaempferol (99.41% purity), luteolin (98% purity), and naringenin (99.55% purity) (all purchased from TargetMol) were dissolved in DMSO to obtain the 10, 30, and 50 mM stock solutions and stored at the temperature of −20 °C until used.

### 4.2. Cell Culture

A human bronchial epithelial 16HBE14σ cell line was obtained from Sigma-Aldrich, Inc. (St. Louis, MO, USA) and grown as previously described [57,58]. Briefly, the cells were grown in Minimal Essential Medium (MEM; MERCK, Darmstadt, Germany) supplemented with 10% fetal bovine serum (FBS; Gibco, ThermoFisher Scientific, Waltham, MA, USA) and antibiotics—100 U/mL penicillin and 100 mg/mL streptomycin (MERCK, Darmstadt, Germany). Cells were cultured on T75 flasks (Nunc, Thermo Fisher Waltham, MA, USA), and passaged upon reaching 70–90% confluence.

### 4.3. Measurement of Cell Viability by MTT Assay

The effect of investigated flavonoids on the metabolic activity of 16HBE14σ cells was determined by the 3-(4,5-dimethylthiazol-2-yl)-2,5-diphenyl-2H-tetrazolium bromide (MTT) assay. Briefly, cells were seeded on 96-well plates (Nunc, Thermo Fisher Waltham, MA, USA) at the density of 5 × 10^4^ cells per well and cultured to reach 100% of confluence. Then, cells were incubated with flavonoids at final concentrations of 10, 30, and 50 μM for 24 h. After 24 h, the medium was removed, and the cells were incubated with 100 μL of fresh medium containing 0.5 mg/mL MTT for 3 h. Then, 100 μL of lysis buffer containing 1:1, ethanol/dimethyl sulfoxide was added to dissolve formazan salt. After 15 min the absorbance was read with a microplate reader at 570 nm (Multiskan SkyHigh TC MD, ThermoFisher Scientific Waltham, MA, USA, serial number: 1600500). The results were expressed as the percentage of alive (metabolically active) cells after 24 h of incubation. Average values were calculated from at least five wells in each group. All experiments were performed with different cell batches and repeated twice. The observed absorbance level results were normalized to results obtained from untreated cell batches in each repetition.

### 4.4. Measurement of Flavonoid Cytotoxicity by Trypan Blue Staining

The cytotoxicity of flavonoids was investigated using Trypan Blue staining. 16HBE14σ cells were incubated with flavonoids at various concentrations for 24, 48, and 72 h. After the incubation, cells were detached with trypsin, resuspended in a culture medium, and incubated with 1:1 medium/Trypan Blue dye for 3 min. Cell survival was measured in a TC20 cell counter (BioRad, Hercules, CA, USA).

### 4.5. Wound Healing Assay

The migration and proliferation of cells were examined by the scratch assay method. 16HBE14σ cells were seeded on 12-well plates (Sarstedt, Numbrecht, Germany) at a density of 2.5 × 10^5^ cells/well and allowed to grow for 48 h to reach 100% of confluence. A small linear scratch was created in the confluent monolayer by scrapping with sterile P100 μL pipette tips. Cells were rinsed with PBS twice to remove all the cellular debris before the addition of fresh cell culture medium without (control) or with supplemented flavonoids at varying concentrations (10, 30, 50 μM). The scratch areas were monitored and photographed by DLTX1080PCMOSHDU2SD camera (DELTA Optical, Minsk Mazowiecki, Poland) in a reversed optical microscope (OLYMPUS IMT-2, Tokyo, Japan) at time 0 h and after 3, 6, and 24 h. Pictures of wound closures were saved in TIFF format by DLT Cam Viewer software (version released on 08/08/2023) and analyzed using Image-J software (1.54k version). The results were presented as a percentage of scratch area at different time points, and the decrease in closed area indicated cell migration and proliferation. The experiments were performed at least in triplicate for each flavonoid.

### 4.6. Epithelial Cell Monolayer Integrity

The effect of flavonoids on cell monolayer integrity was assessed by the measurement of TEER. Briefly, the cells were seeded onto Corning Costar Snapwell inserts (0.45 μM, 1.12 cm^2^ surface area) and grown to form tight and polarized cell monolayers, as described previously [59,60]. Before experiments, cell monolayers were gently rinsed with sterile PBS solution, followed by the addition of fresh culture media. Flavonoids under investigation were then added to the apical solution at final concentrations of 10, 30, and 50 μM and the TEER was measured at 0, 24, 48, and 72 h after treatment by means of EVOM2 voltmeter with STX-2 chopstick electrodes (World Precision Instruments Sarasota, FL, USA). The TEER value of the blanc insert (with medium only) was subtracted from each reading, and the results were then multiplied by the cell layer surface area to express the results in standard units as Ωcm^2^.

### 4.7. Short-Circuit Current Measurements

Among all the flavonoids tested, luteolin at a concentration of 50 μM had the strongest impact on the transepithelial electrical resistance of 16HBE14o- cell monolayers, but at the same time, it almost completely inhibited cell proliferation. To determine whether the observed increase in transepithelial resistance is solely due to the tightening of intercellular junctions or also due to its inhibitory effect on ion channels, we performed short-circuit current measurements on cell monolayers treated with luteolin. The effect of luteolin on short-circuit current was assessed by the Ussing chamber technique. The polarized cell monolayers were treated with luteolin for 72 h and mounted in the Ussing chambers (Easy Mount Chamber system EM-CSYS-2, Physiologic Instruments, Reno, NV, USA). Short-circuit currents were measured with an EVC4000 Multi-Channel V/I Clamp (World Precision Instruments, Sarasota, FL, USA) and recorded and analyzed via an iWorx118 data acquisition system with LabScribe software 4.0 (iWorx, Dover, NH, USA). Chambers were filled with 5 mL on each side. The basolateral chamber was filled with 145 mM NaCl, 3.3 mM K_2_HPO_4_, 10 mM HEPES, 10 mM D-glucose, 1.2 mM MgCl_2,_ and 1.2 mM CaCl_2_ while in the apical solution, NaCl was replaced by 145 mM Na-gluconate to generate transepithelial chloride gradient. Transepithelial voltage was clamped to zero (V_hold_ = 0 mV) using ECV4000 Precision V/I clamp (World Precision Instruments, Sarasota, FL, USA), and the short-circuit current was recorded by an analog-to-digital converter (iworx118, iWorx, Dover, NH, USA) connected to PC with LabScribe 4.0 data acquisition software (iworx, Dover, NH, USA). After stabilizing the short-circuit current, the cells were subsequently treated with the following: 100 μM amiloride (Merck, Darmstad, Germany) apically to inhibit ENaC channel, 10 μM forskolin (Tocris Bioscience, Bristol, UK) to activate CFTR channel, 10 μM CFTR_Inh_-172 (Merck, Darmstadt, Germany) to inhibit CFTR channel, and 100 μM ATP (Merck, Darmstadt, Germany) to investigate purinergic calcium-dependent Cl^−^ secretion. Isc change after stimulation with cAMP agonists and its inhibition by Inh-172 (∆Isc_inh-172_) served as an index of CFTR function.

### 4.8. Data Processing and Statistical Analysis

Acquired numerical data were processed in Excel (Microsoft Office 365) and GraphPadPrism version 4.03 (Dotmatics, Boston, MA, USA). One-way ANOVA was used for statistical analysis. Data were presented as Means ± Standard Deviation (SD). Photographs were analyzed in Image-J software (Java). Figures were prepared in CorelDraw 2023 version 24.3.0.571 (Corel Corporation, Ottawa, CA, USA).

## 5. Conclusions

This study underscores the significant potential of flavonoids—specifically kaempferol, luteolin, and naringenin—in enhancing airway epithelial barrier function. Given the critical role of the respiratory epithelium in protecting the body from environmental threats, including urban particulate matter, these findings highlight how flavonoids can bolster barrier integrity. The distinct effects observed among the flavonoids suggest tailored therapeutic applications in respiratory health, particularly in improving transepithelial electrical resistance, promoting wound healing, and supporting cell proliferation. As respiratory diseases are often linked to epithelial compromise, incorporating flavonoid-rich foods into our diets may offer a proactive approach to maintaining respiratory health. Further research into the mechanisms of action will be essential for developing targeted therapies that harness the benefits of these bioactive compounds.

## Figures and Tables

**Figure 1 ijms-25-11999-f001:**
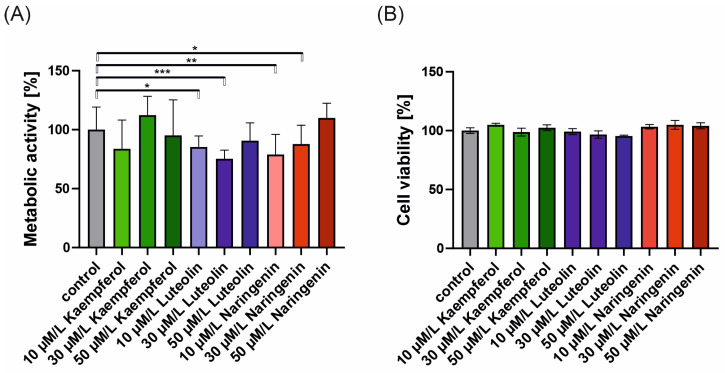
The effect of flavonoids on 16HBE14σ cells metabolic activity and cell viability. (**A**) changes in cell metabolic activity after 24 h incubation with flavonoids (kaempferol, luteolin, naringenin) at various concentrations (10, 30, and 50 μM). (**B**) Cell viability of cells incubated with flavonoids for 24 h was assessed by trypan blue staining. The results were normalized to control and presented as mean ± standard deviation (SD). Statistical analysis was evaluated by one-way ANOVA (* *p* < 0.05, ** *p* < 0.01, *** *p* < 0.001).

**Figure 2 ijms-25-11999-f002:**
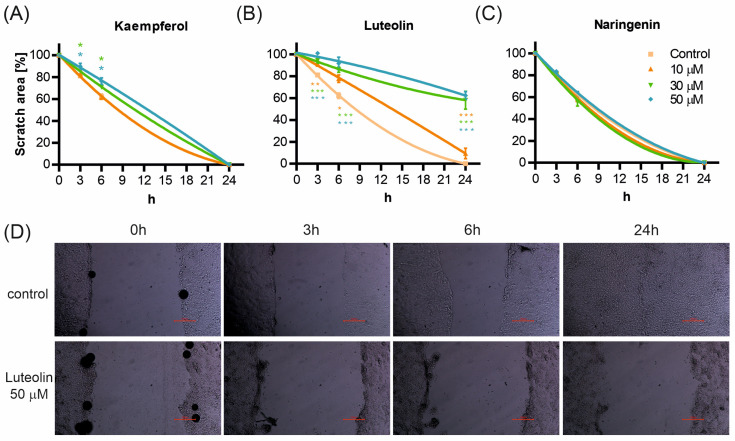
16HBE14σ cell migration in a scratch assay. Quantitative analyses of the migration assays are expressed as percentages relative to the area at time 0 h. after 3, 6, and 24 h after treatment with different concentrations of kaempferol (**A**), luteolin (**B**), and naringenin (**C**) at various concentrations. (**D**) Example images showing scratch closure process of control and 50 μM luteolin treated cells over time. The red scale indicates the distance of 100 μM. The data present mean values of scratch areas ± standard deviations (SD) (*n* = 3). Statistical analysis was assessed by one-way ANOVA (* *p* < 0.05, ** *p* < 0.01, *** *p* < 0.001).

**Figure 3 ijms-25-11999-f003:**
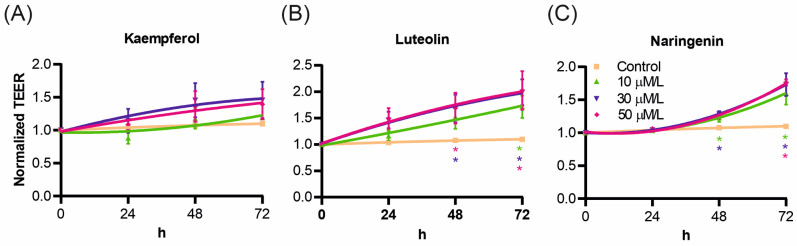
Impact of flavonoids on transepithelial electrical resistance in 16HBE14σ cell monolayers. Tight and polarized cell monolayers were treated apically with kaempferol (**A**), luteolin (**B**), and naringenin (**C**) at concentrations of 10, 30, and 50 μM. TEER was measured after 24, 48, and 72 h of incubation. Results were normalized to the TEER value measured before treatment (time 0 h). Graphs display mean values ± standard deviations (SD). Statistical significance compared to control was determined using a one-way ANOVA (* *p* < 0.05).

**Figure 4 ijms-25-11999-f004:**
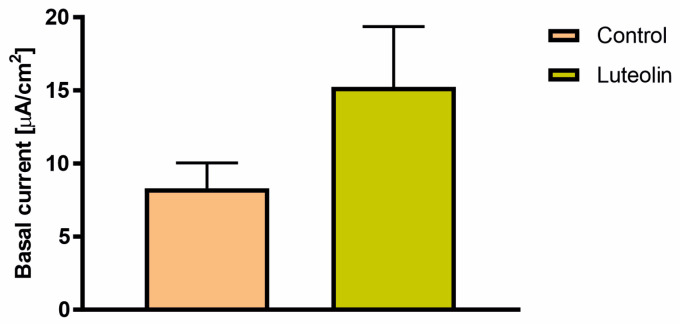
Changes in basal chloride current flowing through 16HBE14σ cell monolayers treated with 50 μM luteolin for 72 h. Chloride current was measured in the Ussing chamber in basolateral to apical chloride gradient. The data are presented as mean ± standard deviations (SD).

**Figure 5 ijms-25-11999-f005:**
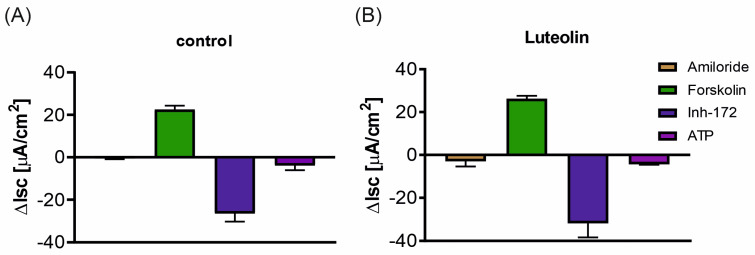
The effect of specific ion channel modulators on short-circuit current. (**A**) short circuit current changes in non-treated 16HBE14σ cell monolayers. (**B**) short circuit current changes in luteolin treated 16HBE14σ cell monolayers. The modulators were added subsequently: amiloride (epithelial sodium channel ENaC inhibitor), forskolin (activator of cAMP-dependent CFTR current), Inh-172 (specific CFTR inhibitor), and ATP (to investigate purinergic calcium-dependent Cl^−^ transport). Data are presented as mean ± standard deviations (SD).

## Data Availability

Data is contained within the article.

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
