# Peer review of "Modulation of the Respiratory Epithelium Physiology by Flavonoids—Insights from 16HBEσcell Model"

_ijms, 2024, doi:10.3390/ijms252211999_

Round 1

Reviewer 1 Report

Comments and Suggestions for Authors

The title does not reflect the manuscript content and needs to be corrected. Consider the types of studies required to evaluate any compounds' therapeutic potential, including flavonoids.
This is a research article, and the abstract, as the short version of the article, needs to include the methodology, main results, and conclusions.
The Introduction is too general and is more appropriate for the review article. Have any previous studies investigated the effects of selected flavonoids on airway epithelium parameters? What is the novelty of this study? What are the research hypotheses?
Are there any data on estimated dietary intakes of kaempferol, luteolin, and naringenin? What are the bioavailability rates of these flavonoids?
What were the criteria for defining tested concentrations of flavonoids? What is IC50 for analyzed flavonoids?
What is the stability of investigated flavonoids in a cell culture medium after 24h, 48h, and 72 hours?
What is the minimum number of replicates needed to evaluate cell viability?

Author Response

Dear Reviewer,

Thank you for your kind review of our manuscript. We are very pleased that you liked it and that you appreciated its scientific value. We are also grateful for pointing out the manuscript's weaknesses, minor errors, and ambiguities. Below, we present answers to your suggestions and questions. We hope that we

The title does not reflect the manuscript content and needs to be corrected. Consider the types of studies required to evaluate any compounds' therapeutic potential, including flavonoids.

Thank you for this suggestion. We rewrote the title of the manuscript.

This is a research article, and the abstract, as the short version of the article, needs to include the methodology, main results, and conclusions.

Thank you very much. In accordance with your suggestion, the abstract has been rewritten

The Introduction is too general and is more appropriate for the review article. Have any previous studies investigated the effects of selected flavonoids on airway epithelium parameters? What is the novelty of this study? What are the research hypotheses?

Thank you for this comment, the introduction was changed. We hope that the revised version of the manuscript will satisfy your expectations.

Are there any data on estimated dietary intakes of kaempferol, luteolin, and naringenin? What are the bioavailability rates of these flavonoids?

Thank you for bringing valuable attention to the issues related to the intake and bioavailability of these flavonoids.

The intake of flavonoids like kaempferol, luteolin, and naringenin varies based on dietary habits, regional diets, and the consumption of specific flavonoid-rich foods.

  1. Kaempferol: Chun et al. (https://doi.org/10.1093/jn/137.5.1244) estimated that the average intake of kaempferol in U.S. adults ranges from 1-10 mg/day, with higher intakes observed in individuals consuming more green vegetables and teas. Similarly, Mullie et al. (https://doi.org/10.1080/09687630701539293) reported a kaempferol intake of 4.6 ± 4.2 mg/day in Belgian populations, indicating the importance of diet composition in flavonoid intake levels.
  2. Luteolin: Its intake is typically lower than other flavonoids due to the lesser consumption of these herbs in Western diets. Chun et al. (https://doi.org/10.1093/jn/137.5.1244) reported an estimated intake of 0.5-2 mg/day for luteolin among U.S. adults, highlighting the need for diverse vegetable and herb consumption to increase intake.
  3. Naringenin: Studies such as those by Zamora-Ros et al. (https://doi.org/10.1017/S000711451100239X ) have reported intakes ranging from 2-15 mg/day, with significant variation based on citrus fruit consumption in different European populations. Mullie et al. (https://doi.org/10.1080/09687630701539293) also reported that naringenin intake could reach 5.1 mg/day among those with high citrus consumption.

The bioavailability of these flavonoids varies significantly due to factors such as solubility, metabolism, and food matrix effects, impacting their physiological efficacy.

  1. Kaempferol: Kaempferol is known for its low bioavailability, with absorption rates ranging from 2-15%. Calderon-Montano et al. (https://doi.org/10.2174/138955711795305335) discussed the challenges of kaempferol absorption due to its poor water solubility and rapid hepatic metabolism.
  2. Luteolin: Luteolin’s bioavailability ranges from 5-20%, with variations depending on the dietary matrix. Its extensive metabolism in the intestines and liver results in glucuronide and sulfate conjugates, which may alter its biological effects Chun et al. (https://doi.org/10.1093/jn/137.5.1244).
  3. Naringenin: Naringenin exhibits higher bioavailability than kaempferol and luteolin, with 15-60% reported rates. Erlund et al. (1038/sj.ejcn.1601409 ) highlighted that naringenin’s absorption is significantly enhanced when consumed with citrus fruit matrices, as they contain other compounds that facilitate uptake.

What were the criteria for defining tested concentrations of flavonoids? What is IC50 for analyzed flavonoids?

The criteria for defining tested concentrations of flavonoids were established based on our previous work (https://doi.org/10.1111/exd.13903, https://doi.org/10.3390/molecules25133010, https://doi.org/10.1016/j.biopha.2021.112039, https://doi.org/10.1016/j.mito.2022.04.005, https://doi.org/10.3390/ijms24010638, https://doi.org/10.1093/jpp/rgac093, https://doi.org/10.3390/antiox11101892). In the long period of our research team's work, it was possible to indicate that concentrations of about 10 µM are effective in experiments based on single proteins (e.g., patch-clamp). Using flavonoids in biochemical studies is beneficial at the higher ranges of 50-100 µM.

What is the stability of investigated flavonoids in a cell culture medium after 24h, 48h, and 72 hours?

We would like to thank the Reviewer for highlighting the issue of flavonoid stability in in vitro systems. We recognize that flavonoid stability in cell culture medium can significantly influence their efficacy in experimental settings. Several studies report a decrease in flavonoid stability in cell culture media (https://10.2478/v10222-011-0048-y, https://doi.org/10.2298/JSC150706092W, https://doi.org/10.1021/jf505514d, https://doi.org/10.1016/j.abb.2010.06.012). Xiao and Hogger (https://10.1021/jf505514d) demonstrated that the stability of flavonoids from our study decreases in DMEM cell culture medium, with naringenin being the most stable, followed by luteolin, and kaempferol the least stable. In contrast, Fang et al. (https://10.3390/nu9121301) reported higher stability of the flavonoids used in our study in Hank’s buffer. MEM that was used as a cell culture medium differs from DMEM by different glucose (that was also shown to affect the flavonoids stability) and bicarbonate concentration stabilizing pH of the medium (the pH was described by the others as a key factor influencing flavonoids stability) In our experiments, we implemented steps to mitigate factors affecting flavonoid stability: flavonoids were dissolved in DMSO, protected from light exposure, and the cell culture medium contained proteins (from fetal bovine serum), which may help stabilize flavonoids, albeit with a potential reduction in bioavailability. Despite the potential decrease in flavonoid concentration over time, the effects on the parameters we studied—proliferation, transepithelial electrical resistance, and ionic currents—remained evident even after 72 hours, suggesting that the treatment exerts sustained effects on 16HBE14o- cells. Long term effects of very unstable flavonoids (such as quercitin) observed in vivo are known in the literature and called flavonoid paradox (https://doi.org/10.1002/jsfa.5697). It should also be noted that the degraded or metabolized polyphenols may exhibit bioactivity distinct from that of the parent compounds.

What is the minimum number of replicates needed to evaluate cell viability?

Minimum number of replicates normally found in biological research articles equals 3. In our experiments, we used 5 replicates for each compound and 2 technical repeats. In our opinion, it is enough to assess the compound's effect on cell viability.

Reviewer 2 Report

Comments and Suggestions for Authors

The experimental work reported here is relatively simple, but the paper is well-thought, well articulated and clear in its facts. The study limitations are discussed in an appropriate manner. We have the following minor comments to make:

After reading the paper, I have remained with the impression that the title is a little bit misleading: the authors have not investigated the effect of flavonoids “in the physiology of the  respiratory epithelium”, but rather the effect of flavonoids on a human bronchial epithelial cell line. From this, one can speculate about the impact of flavonoids on the respiratory epithelium, but the title should faithfully reflect what was investigated. The same holds true for the sentence on lines 162-163.

Line 18 – please correct “flawon” to “flavone” and “flavanon” to “flavanone”.

Line 50-51: “Recently, more than 10 000 different flavonoid compounds have been  extracted, isolated and identified…” The text, as currently phrased, suggests that all 10,000 flavonoid compounds have been identified recently, whereas they were identified across time, up to a recent time point. This should be corrected.

Line 55: from the enumeration the class of “flavanonols” is absent, although it definitely does exist.

Line 238: “Then, cells were incubated with  flavonoids at a final concentrations of 10, 30, and 50 μM for 24h.” What volume of the flavonoid solutions was used?

Lines 281 – 297: please clarify here why only luteolin was used in this experiment.

With respect to Figure 1A: it is not clear, and the authors do not discuss: why is there lack of any clear dose-response relationship (except for naringenin)?

The discussion should include (in our view) a paragraph on the potential applications that the experimental results suggest, particularly for luteolin (as the others do not seem to have a large impact on epithelial cells).

Author Response

Dear Reviewer,

Thank you for your kind review of our manuscript. We are very pleased that you liked it and that you appreciated its scientific value. We are also grateful for pointing out the manuscript's weaknesses, minor errors, and ambiguities. Below, we present answers to your suggestions and questions. We hope that we have made our statements exhaustively and that they will dispel your doubts.

The experimental work reported here is relatively simple, but the paper is well-thought, well articulated and clear in its facts. The study limitations are discussed in an appropriate manner.

Thank you for your kind comment.

We have the following minor comments to make:

After reading the paper, I have remained with the impression that the title is a little bit misleading: the authors have not investigated the effect of flavonoids “in the physiology of the respiratory epithelium”, but rather the effect of flavonoids on a human bronchial epithelial cell line. From this, one can speculate about the impact of flavonoids on the respiratory epithelium, but the title should faithfully reflect what was investigated. The same holds true for the sentence on lines 162-163.

We fully agree with this comment. The title has been rewritten. Additionally, changes in lines 162-163 were made.

Line 18 – please correct “flawon” to “flavone” and “flavanon” to “flavanone”.

Thank you for finding this mistake, the names were corrected in the main text of revised manuscript.

Line 50-51: “Recently, more than 10 000 different flavonoid compounds have been  extracted, isolated and identified…” The text, as currently phrased, suggests that all 10,000 flavonoid compounds have been identified recently, whereas they were identified across time, up to a recent time point. This should be corrected.

Thank you for this comment. We have changed the sentence to: “So far, more than 10,000 different flavonoid compounds have been found, extracted, isolated, and identified”. We believe that this sentence is now not misleading.

Line 55: from the enumeration the class of “flavanonols” is absent, although it definitely does exist.

Thank you for highlighting this issue. Indeed, the literature presents different approaches to the classification of flavonoids. When writing the article, we used the six-group classification. However, we agree with the reviewer and fully acknowledge the existence of the flavanonol group. We changed the sentence in the revised manuscript and added a new reference indicating a seven-group classification that includes the flavanonols class.

Depending on the chemical structure, degree of saturation, and the oxidation of the carbon ring, flavonoids are classified into one of seven significant subclasses such as flavanones, flavones, isoflavones, flavonols, flavanonols, flavanols, and anthocyanins [18].

Line 238: “Then, cells were incubated with  flavonoids at a final concentrations of 10, 30, and 50 μM for 24h.” What volume of the flavonoid solutions was used?

In the experiments, we used stock solutions at the concentration 1000x higher than the final concentration used in the experiments. Thus we add 1 µL of appropriate stock solution per 1 mL of experimental buffer/medium

Lines 281 – 297: please clarify here why only luteolin was used in this experiment.

The selection of luteolin for short-circuit current measurements was explained in the Results section. However, for greater clarity, we have extended the appropriate “Materials and methods” section with the additional sentences:

Among all the flavonoids tested, luteolin at a concentration of 50 µM had the strongest impact on the transepithelial electrical resistance of 16HBE14o- cell monolayers, but at the same time, it almost completely inhibited cell proliferation. To determine whether the observed increase in transepithelial resistance is solely due to the tightening of intercellular junctions or also due to its inhibitory effect on ion channels, we performed short-circuit current measurements on cell monolayers treated with luteolin.With respect to Figure 1A: it is not clear, and the authors do not discuss: why is there lack of any clear dose-response relationship (except for naringenin)?

Thank you for raising this issue; however, the answer is not straightforward. We routinely use the MTT assay to examine the effects of various compounds on cellular metabolic activity, and for most of them, we observe a dose-response relationship. Its lack in this study might be caused by several reasons:

  • Limitations of the MTT assay as flavonoids were described to reduce MTT even in the absence of the cells (https://10.1007/s10616-007-9057-4).
  • The drop in MTT was observed for some flavonoids used in low concentrations, with the increase at higher concentrations (a reversed profile of metabolic activity at a concentration of 25 µM was observed by Somayaji and Shastry (https://10.9734/JPRI/2021/v33i49A33305).
  • Medium type and serum used in our study might also affect the MTT results. As shown by Talorete et al. (https://10.1007/s10616-007-9057-4), the effect of some flavonoids on MTT results did not show the time and concentration-dependent trend.
  • Differences in membrane permeability and cellular accumulation of investigated flavonoids (https://10.3390/nu9121301).
  • Different stability of investigated flavonoids in cell culture medium. Degraded or metabolized polyphenols may exhibit bioactivity distinct from that of the parent compounds, i.e., leading to activation or inhibition of intracellular enzymes.

Although there is no clear dose-response relationship, we did not observe statistically significant changes in cell metabolic activity across different concentrations of each flavonoid. To determine whether these changes were due to the cytotoxicity of the compounds used, we performed trypan blue staining, which confirmed no cytotoxic effect of the investigated compounds.

The discussion should include (in our view) a paragraph on the potential applications that the experimental results suggest, particularly for luteolin (as the others do not seem to have a large impact on epithelial cells).

Thank you for this suggestion. A paragraph was added to the revised version of the manuscript.

Reviewer 3 Report

Comments and Suggestions for Authors

I reviewed the manuscript entitled the therapeutic potential of flavonoids in the physiology of the respiratory epithelium.

 I agree to accept this manuscript after major revision. 

1) Keywords: airway epithelium; As the first keyword, the first letter of its first word should be capitalized. transepithelial electrical resistance (TEER); Do not use abbreviations in keywords, therefore remove (TEER).

2) chronic obstructive pulmonary disease (COPD), and cystic fibrosis (CF), These two abbreviations, the former appears twice and the latter only appears once. The general principle of using abbreviations is that they are only necessary when they appear three or more times. Otherwise, too many abbreviations will confuse readers. Please recheck and revise similar issues throughout the text. Meanwhile, CFTR has been used many times, but it did not have its full name when it first appeared. Please add it.

3) 2.1. Flavonoids affect cell metabolic activity but not cell viability. The first letter of each actual word in the secondary title needs to be capitalized, check and modify the entire text.

4) 24h incubation of cells with kaempferol led to non-significant changes…  24h should change to 24 h. There needs to be a space between numbers and international units. There is no need for spaces between numbers and% and oC.check and modify the entire text.

5) The results of the MTT assay and trypan blue staining are presented in Figure 1 A, Figure 1 A should change to Figure 1A.

6) The trypan blue staining assess the possible cytotoxicity of selected flavonoids. As presented in Figure 1B, none of the flavonoids tested had a cytotoxic effect on 16HBE14σ cells.  Why choose 16HBE14 σ cells? What is the reason? Why not screen more cells?

7) Figure 1.  *p<0.05, **p<0.01, ***p<0.001. When it comes to statistics, p should be italicized.

8) Figure 2. 10M/ml should change to10 M/mL, But the descriptions in the article are all 10 μ M, which one is correct? Please verify and make revisions by the author. The unit of the horizontal axis should also be changed to h, and international units must be used in scientific papers instead of words.

9) 2.3. Flavonoids affect the transepithelial electrical resistance (TEER), Abbreviations have already been defined here, you can use them directly afterwards.

10) EasyMount Ussing chamber system (Physiologic Instruments, USA), Is there a specific model for this instrument? If possible, please add it.

11) Apigenin, daidzein, kaempferol, luteolin, morin and naringenin (all purchased from TargetMol) were dissolved in DMSO…The purity of these compounds needs to be provided.

12) The effect of investigated flavonoids on the metabolic activity of 16HBE14σ cells was determined by the MTT (3-(4,5-dimethylthiazol-2-yl)-2,5-diphenyl-2H-tetrazolium bromide) assay. Wrote it backwards, it should be 3-(4,5-dimethylthiazol-2-yl)-2,5-diphenyl-2H-tetrazolium bromide (MTT), And abbreviations have already appeared in the previous text, so the full name and abbreviation should be placed where they first appear.

13) Multiskan SkyHigh TC MD, Add the specific model.

14) 4.8. Data processing and statistical analysis, Did the author use one-way ANOVA? Specific methods of use should be listed.

15) Future prospects should be added to the conclusion section.

16) The primary question tackled by the research is addressed, but it is necessary to address excessive errors in details and insufficient discussion by incorporating a discussion section that delves deeper into the experimental results.

17) The respiratory epithelium is a vital barrier between the body's internal and external environments. Its dysfunction is linked to various diseases, impacting global health. Flavonoids, natural compounds with antioxidant and anti-inflammatory properties, have been used for centuries in treatment. Evidence suggests they can enhance airway epithelial barrier integrity. This study investigated the effects of kaempferol, luteolin, and naringenin on human bronchial epithelial cells' transepithelial electrical resistance, ionic currents, wound-healing, and proliferation. Results show that these flavonoids have distinct effects despite similar structures. Our study highlights their diverse impacts on airway barrier function, emphasizing further exploration for respiratory health applications.

18) The other published materials on this topic mainly focus on the treatments that target airway barrier dysfunction, including anti-inflammatory therapies to reduce inflammation, interventions to enhance epithelial barrier integrity, antioxidant use to mitigate oxidative stress, and mucolytics to improve mucociliary clearance. The objective of this study was to investigate the impact of selected flavonoids such as kaempferol, luteolin, and naringenin on various aspects of airway epithelium parameters such as transepithelial electrical resistance, ionic currents, wound-healing properties, and proliferation.

19) The conclusion is consistent with the evidence and arguments provided. All the main questions raised by the author have been resolved.

20) I have read all the references and found two issues. Ref 14, 1–9 is wrong, it should change to 5445291. Ref 44,  Missing article number, I retrieved it, it should be 1881, please add it.

Author Response

Dear Reviewer,

Thank you for your kind review of our manuscript. We are very pleased that you liked it and that you appreciated its scientific value. We are also grateful for pointing out the manuscript's weaknesses, minor errors, and ambiguities. Below, we present answers to your suggestions and questions. We hope that we have made our statements exhaustively and that they will dispel your doubts.

I reviewed the manuscript entitled the therapeutic potential of flavonoids in the physiology of the respiratory epithelium.

I agree to accept this manuscript after major revision. 

1) Keywords: airway epithelium; As the first keyword, the first letter of its first word should be capitalized. transepithelial electrical resistance (TEER); Do not use abbreviations in keywords, therefore remove (TEER).

We fully agree with the Reviewer. Therefore, we deleted the TEER abbreviation from the keywords and started the first keyword with a capital letter.

2) chronic obstructive pulmonary disease (COPD), and cystic fibrosis (CF), These two abbreviations, the former appears twice and the latter only appears once. The general principle of using abbreviations is that they are only necessary when they appear three or more times. Otherwise, too many abbreviations will confuse readers. Please recheck and revise similar issues throughout the text. Meanwhile, CFTR has been used many times, but it did not have its full name when it first appeared. Please add it.

We are very thankful for the in-depth review of our manuscript. We addressed all the suggestions and corrected abbreviations as suggested.

3) 2.1. Flavonoids affect cell metabolic activity but not cell viability. The first letter of each actual word in the secondary title needs to be capitalized, check and modify the entire text.

We capitalized the first letter of each actual word in the secondary title, and we carefully checked and modified the entire text as suggested.

4) 24h incubation of cells with kaempferol led to non-significant changes…  24h should change to 24 h. There needs to be a space between numbers and international units. There is no need for spaces between numbers and% and oC.check and modify the entire text.

Thank you for your comments. We modified the manuscript body.

5) The results of the MTT assay and trypan blue staining are presented in Figure 1 A, Figure 1 A should change to Figure 1A.

Thank you. The change has been made in the manuscript.

6) The trypan blue staining assess the possible cytotoxicity of selected flavonoids. As presented in Figure 1B, none of the flavonoids tested had a cytotoxic effect on 16HBE14σ cells.  Why choose 16HBE14 σ cells? What is the reason? Why not screen more cells?

16HBE14σ- is a human bronchial epithelial cell line isolated from a 1-year-old male heart-lung patient and immortalized with the origin of replication-defective SV40 plasmid. The cell line represents characteristic features of normal differentiated bronchial epithelial cells, including a cobblestone morphology, cytokeratin expression, and the ability to form tight junctions, and was described as a perfect model for measuring ion transport. When grown with an air/liquid interface, cilia can be detected. In contrast to most other respiratory cell lines, this one expresses high levels of cystic fibrosis transmembrane conductance regulator (CFTR) mRNA and protein. The expression of CFTR is correlated with cAMP-dependent Cl- conductance in various cells, including 16HE14σ- epithelial cells.

7) Figure 1.  *p<0.05, **p<0.01, ***p<0.001. When it comes to statistics, p should be italicized.

Thank you for the indication. We have changed the font to italics.

8) Figure 2. 10M/ml should change to10 M/mL, But the descriptions in the article are all 10 μ M, which one is correct? Please verify and make revisions by the author. The unit of the horizontal axis should also be changed to h, and international units must be used in scientific papers instead of words.

Thanks, this is a very accurate remark, the units have been changed.

9) 2.3. Flavonoids affect the transepithelial electrical resistance (TEER), Abbreviations have already been defined here, you can use them directly afterwards.

We agree with the Reviewer; we used the first TEER abbreviation explanation in the Introduction. We addressed the suggestions and corrected them as proposed in this subsection.

10) EasyMount Ussing chamber system (Physiologic Instruments, USA), Is there a specific model for this instrument? If possible, please add it.

A revised version of the manuscript contains the full description of the Ussing chamber system used in the experiments.

“The polarized cell monolayers were treated with luteolin for 72 h and mounted in the Ussing chambers (Easy Mount Chamber system EM-CSYS-2, Physiologic Instruments, NV, USA). Short-circuit currents were measured with an EVC4000 Multi-Channel V/I Clamp (World Precision Instruments, FL, USA) and recorded and analyzed via an iWorx118 data acquisition system with LabScribe software (iWorx, NH, USA).”

11) Apigenin, daidzein, kaempferol, luteolin, morin, and naringenin (all purchased from TargetMol) were dissolved in DMSO…The purity of these compounds needs to be provided.

The purity of selected flavonoids was added in the revised version of the manuscript.

12) The effect of investigated flavonoids on the metabolic activity of 16HBE14σ cells was determined by the MTT (3-(4,5-dimethylthiazol-2-yl)-2,5-diphenyl-2H-tetrazolium bromide) assay. Wrote it backwards, it should be 3-(4,5-dimethylthiazol-2-yl)-2,5-diphenyl-2H-tetrazolium bromide (MTT), And abbreviations have already appeared in the previous text, so the full name and abbreviation should be placed where they first appear.

We have corrected the MTT explanation according to the Reviewer's suggestion.

13) Multiskan SkyHigh TC MD, Add the specific model.

A specific description of the equipment has been added.

14) 4.8. Data processing and statistical analysis, Did the author use one-way ANOVA? Specific methods of use should be listed.

Thank you for indicating it; yes, we did. Information has been added to the revised manuscript.

15) Future prospects should be added to the conclusion section.

Thank you. We added appropriate text in the conclusion section.

16) The primary question tackled by the research is addressed, but it is necessary to address excessive errors in details and insufficient discussion by incorporating a discussion section that delves deeper into the experimental results.

The discussion section has been rewritten. We hope that the revised version of the manuscript will satisfy your expectations.

17) The respiratory epithelium is a vital barrier between the body's internal and external environments. Its dysfunction is linked to various diseases, impacting global health. Flavonoids, natural compounds with antioxidant and anti-inflammatory properties, have been used for centuries in treatment. Evidence suggests they can enhance airway epithelial barrier integrity. This study investigated the effects of kaempferol, luteolin, and naringenin on human bronchial epithelial cells' transepithelial electrical resistance, ionic currents, wound-healing, and proliferation. Results show that these flavonoids have distinct effects despite similar structures. Our study highlights their diverse impacts on airway barrier function, emphasizing further exploration for respiratory health applications.

18) The other published materials on this topic mainly focus on the treatments that target airway barrier dysfunction, including anti-inflammatory therapies to reduce inflammation, interventions to enhance epithelial barrier integrity, antioxidant use to mitigate oxidative stress, and mucolytics to improve mucociliary clearance. The objective of this study was to investigate the impact of selected flavonoids such as kaempferol, luteolin, and naringenin on various aspects of airway epithelium parameters such as transepithelial electrical resistance, ionic currents, wound-healing properties, and proliferation.

Thank you for your comments. We have explored the topics in the Introduction and Discussion session.

19) The conclusion is consistent with the evidence and arguments provided. All the main questions raised by the author have been resolved.

Thank you for your kind commentary.

20) I have read all the references and found two issues. Ref 14, 1–9 is wrong, it should change to 5445291. Ref 44,  Missing article number, I retrieved it, it should be 1881, please add it.

Thank you for finding the issues with References, the corrections have been made in the revised version of the manuscript.

Round 2

Reviewer 1 Report

Comments and Suggestions for Authors

The Authors have addressed all of my concerns about the original manuscript.

Author Response

Dear Reviewer,

Thank you.

Reviewer 3 Report

Comments and Suggestions for Authors

The author has revised the article as per my request and explained my doubts, therefore I agree to accept it.

Author Response

Dear Reviewer,

Thank you.